# Evidence of a Proximity Effect in a (AgI)*_x_* − C_(1−x)_ Mixture Using a Simulation Model Based on Random Variable Theory

**DOI:** 10.3390/molecules29112491

**Published:** 2024-05-24

**Authors:** Hernando Correa, Diego Peña Lara, Edgar Mosquera-Vargas

**Affiliations:** 1Instituto Interdisciplinario de las Ciencias, Universidad del Quindío, Armenia 630004, Colombia; hecorrea@uniquindio.edu.co; 2Grupo de Transiciones de Fases y Materiales Funcionales, Departamento de Física, Cali 760032, Colombia; edgar.mosquera@correounivalle.edu.co; 3Centro de Excelencia en Nuevos Materiales (CENM), Universidad del Valle, Cali 760032, Colombia

**Keywords:** silver iodide, carbon, phenomenological models, transition temperature, ionic conductivity, probability distribution

## Abstract

Silver iodide is a prototype compound of superionic conductors that allows ions to flow through its structure. It exhibits a first-order phase transition at 420 K, characterized by an abrupt change in its ionic conductivity behavior, and above this temperature, its ionic conductivity increases by more than three orders of magnitude. Introducing small concentrations of carbon into the silver iodide structure produces a new material with a mixed conductivity (ionic and electronic) that increases with increasing temperature. In this work, we report the experimental results of the ionic conductivity as a function of the reciprocal temperature for the (AgI)*_x_* − C_(1−x)_ mixture at low carbon concentrations (*x* = 0.99, 0.98, and 0.97). The ionic conductivity behavior as a function of reciprocal temperature was well fitted using a phenomenological model based on a random variable theory with a probability distribution function for the carriers. The experimental data show a proximity effect between the C and AgI phases. As a consequence of this proximity behavior, carbon concentration or temperature can control the conductivity of the (AgI)*_x_* − C_(1−x)_ mixture.

## 1. Introduction

Superionic conductors allow for the movement of ions through their structure and exhibit unusually high ionic conductivity values (similar to liquid electrolytes). AgI is a superionic compound whose high conductivity was discovered in 1914 [1] and has been extensively studied experimentally [2,3,4,5,6,7], theoretically [8,9,10,11,12,13,14,15], and with computer simulations [16,17,18,19,20,21,22]. At atmospheric pressure, AgI is polymorphic [23], with phases designated gamma, beta, and alpha (γ, β, and α), which become accessible with increasing temperature T. At room temperature, the γ phase [24] is thermodynamically meta-stable with a zinc blende structure, and the β phase [25] is stable with a wurtzite structure. At the transition temperature *T_t_* = 420 K, AgI undergoes a first-order phase transition from β-AgI to α-AgI [26]. The α phase [27] has high ionic conductivity (σ∼1 S/cm) [28], comparable to the ionic conductors in the liquid phase; for this reason, the AgI compound is considered a superionic conductor [29].

The high ionic conductivity, from 420 K to its melting point at 828 K, has been explained by the disorder of silver ions over 42 available and randomly distributed sites in a crystal structure where the iodide ions are in a fixed sublattice and the silver ions behave in a liquid-like sublattice [30]. Silver iodide has been proposed and used as a solid electrolyte for sensors, electrochemical cells, rechargeable cells, and solid-state batteries [31,32]. These solid electrolytes offer the outstanding advantages of miniaturization, high mechanical stability, rechargeable life, and high-temperature performance [33].

Experimentally, silver iodide has 42 crystallographic sites available for the mobility of Ag ions. According to the results of X-ray diffraction experiments [2], the neutron scattering technique [3], and extended X-ray absorption fine structure investigations [4], the unoccupied sites are octahedral. At the same time, the Ag ions are distributed over the tetrahedral sites.

Several phenomenological [8,9,10,11] and theoretical [12,13,14,15] models have been proposed to explain the basic transport mechanism of silver ions in this case. The diffusion of silver ions is assumed to be hopping [34].

Computational simulations are required to provide a microscopic perspective to understand this system’s nature of silver ion motion and first-order phase transition behavior [19]. Through molecular dynamics (MD), static and dynamic simulations can be used to study these systems [16,17,18,20,21,22].

Carbon has different structural forms called allotropes or polymorphs: fullerene, diamond, graphite, and others [35,36]. These polymorphic forms differ in how the carbon atoms are arranged in the crystalline forms. Fullerenes are structures with 60 carbon atoms arranged in pentagons and hexagons. A diamond is a three-dimensional crystalline structure of carbon atoms, while graphite is a two-dimensional lattice. Carbon is a naturally occurring form of crystalline carbon. The carbon atoms are arranged in a series of stacked parallel layer planes in the X-ray structural analysis of carbon at room temperature and atmospheric pressure. Each atom is bonded to three neighbors at a distance of 1.415 Å and 3.348 Å as the separation between planes [35,37].

The (AgI)*_x_* − C_(1−x)_ mixture is non-conductive at room temperature. The material resulting from the mixture of AgI and carbon is a composite in which each compound retains its physicochemical identity. However, in our particular case, the two compounds are electrostatically correlated by the electrical charge of each component. With increasing temperature, the mobility of the positive silver ions increases, resulting in increased conductivity due to the thermal excitation of the Ag ions interacting electrostatically with the electrons of the carbon bands. At low carbon concentrations, the silver ion concentration dominates the conductivity. When the carbon concentration is significant, the electrons dominate, so the jump probability model cannot fit the conductivity curves as a function of temperature. As the temperature increases, the electrostatic interaction between the electrons of the carbon bands and the silver ions decreases, allowing for a significant amount of silver ions to participate in the conduction process. This work will not consider studying the interaction mechanism between electrons and silver ions as the temperature increases.

The (AgI)*_x_* − C_(1−x)_ mixture has potential use in technology due to its electron and ion conduction properties. When the AgI and C are separated, each shows its particular conductivity behavior, carbon as a semiconductor and AgI as a superionic conductor. The proximity between the AgI and C produces a novel conductivity behavior. Mixing carbon and AgI, a new compound whose conductivity is due to electrons and ions, is obtained. This compound has the property that the conductivity is very low at room temperature, lower than that of pure AgI. Increasing the temperature, the mixture conductivity increases. When the mixture reaches 100 °C, its conductivity is comparable to that of pure carbon. Due to the proximity between the carbon bands and the silver ions, the electrostatic interaction between the semiconductor bands and the Ag cations produces the observed effect on the conductivity. No theoretical aspects have been considered in this work.

The scientific novelty of the study lies in obtaining a carbon-doped ionic compound with mixed (electronic and ionic) temperature-dependent conductivity, which can be controlled by the carbon concentration and has excellent potential for technological applications. The conductivity behavior and the sharp jump during the first-order phase transition of the mixed compound are fitted by a phenomenological model based on the carrier density and the probability distribution of the carriers. The conductivity behavior of AgI ceases to be the Arrhenius type when carbon is added, as demonstrated by experimental evidence. The results show that the non-Arrhenius conductivity fit is suitable for low carbon concentrations but not for high concentrations, where electronic conductivity dominates. The mechanism of conductivity implied by the model is ion hopping, which is not true at high carbon concentrations.

This work is organized as follows: Section 2 presents the results obtained for cases with maximum probability (or p = 1) and having a probability distribution function (or 0 < *p* ≤ 1). Section 3 and Section 4 discuss the results obtained for the Huberman *p* = 1) and random variable theory (0 < *p* ≤1) models, respectively, and Section 5 concludes this work.

## 2. Results

To fit the conductivity behavior, we assume the following interpretations for the variational parameters in Equation (3): η represents the charge carrier concentration or n in Equation (2), where μ is the mobility and τ is related to temperature T (proportional to the inverse of *T*). The roots of Equation (7), which minimize Equation (3), give the best-fitting curve for both the conductivity behavior and the respective abrupt change at *T_t_*. The frequency information is contained in Γ, and the probability distribution is driven by *p*.

Figure 1 displays the behavior of the DC conductivity with the inverse of temperature for (AgI)*_x_* − C_(1−x)_ with different carbon concentrations: pure AgI or 1.00 (black circle), 0.99 (cyan circle), 0.98 (blue circle), 0.97 (green circle), and 0.94 (red circle) in the temperature range from 220 K to 416 K for pure AgI and from 320 K to 413 K for the different concentrations.

Figure 2 exhibits that the variation of the DC conductivity with reciprocal temperature for (AgI)*_x_* − C_(1−x)_ data are fitted according to Equation (7) for the carbon concentrations *x* = 0.94 (red circle), *x* = 0.97 (green circle), *x* = 0.98 (blue circle), and *x* = 0.99 (cyan circle). This behavior is non-Arrhenius for all carbon concentrations and temperatures ranging from 320 K to 413 K. Note that at concentration 0.94, the model only adjusts the abrupt change in conductivity. For the pure silver iodide or *x* = 1.00 (black circle), the behavior is Arrhenius (represented by means of the black line) for temperatures ranging from 220 K to 416 K.

Figure 3 illustrates the behavior of *U*_1_/k_B_*T* dependence with the probability for a low carbon concentration (*x* = 0.98).

## 3. Discussion

Figure 1 shows that the transition temperature for pure silver iodide (black circle) is 416 K. In contrast, for concentrations *x* = 0.99 (cyan circle), 0.98 (blue circle), 0.97 (green circle), and 0.94 (red circle), the transition occurs at 413 K. This variation in the transition temperature value is attributed to the presence of carbon, which absorbs some of the thermal energy by delaying the occurrence of the transition. Note how the magnitude of the abrupt jump in conductivity decreases with increasing carbon concentration according to the proposed model. The dashed line represents the best fit to the abrupt change in ionic conductivity for *x* = 0.98 using Equation (4). It shows an excellent fit to the abrupt change in conductivity when the participation of all carriers or *p* = 1 is considered. The values used for the fit parameters are Γ = 0.73, χ = 1.25, and *p* = 1.

In Figure 2, the inverse dependence of temperature (1000/*T*) versus DC conductivity [ln(σ)] is plotted for the mixture of (AgI)*_x_* − C_(1−x)_ during the first cooling run from 320 K to 413 K. The values are shown for *x* = 0.99, 0.98, 0.97, and 0.94. This figure shows the best fit with the random variable theory model. The plot also shows the abrupt change in DC σ at about *T_t_* = 413 K. The best-fitting parameters are χ = 1.3360, Γ = 0.6289, $\tau_{t}$ = 5.5327, and $p_{c}$ = 0.87. The slope of $\ln\left(\sigma \right)$] as a function of 1000/T is interpreted as the activation energy *E*_act_, which is not constant as T increases. These energies for the different concentrations studied are associated with transport processes and decrease with increasing concentrations. This analysis suggests that the *E*_ac_, associated with charge transport in the (AgI)*_x_* − C_(1−x)_ mixture, is sensitive to carbon concentrations and temperature: it decreases when increasing both the concentration and temperature. Adjustments for carbon concentrations (*x* = 0.99, 0.98, 0.97, and 0.94) in the temperature range of 300 K to 320 K show that the correlation between silver ions in AgI and the electrons of the carbon is dominated by silver due to the jumping mechanism model. For concentration *x* = 0.94, this model fits only the abrupt jump in conductivity. The predicted behavior is a bit far from the experimental data, partly due to the interaction between the silver ions and the electrons in the carbon layers, which is not considered in the model.

In Figure 3, the number of charge carriers (silver ions) increases with increasing temperature. Near the transition temperature, almost all are present, i.e., the probability p is between $p_{c}$ and 1. At *T* = *T_t_*, all ions driving the first-order transition are present, i.e., p = 1. The behavior of *p* as a function of τ is a decreasing exponential curve and was adjusted with the following equation:(1)$$p\left(\tau \right)=a\;\ln\left(-b\tau \right)$$
where a = 8.35 y and b = 0.94. The jumping mechanism only makes sense until just before *T_t_* since at temperatures equal to or higher than *T_t_*, a new transport mechanism takes place since one has the superionic phase, which still needs to be better understood.

## 4. Materials and Methods

The (AgI)*_x_* − C_(1−x)_ mixture was prepared from 99.9% purity carbon powder (Alfa Aesar EINECS: 231-955-3, product of United States, Ward Hill, MA, and 99.9% purity AgI powder (Alfa Aesar) EINECS: 232-038-0, CAS 7783-96-2, product of Lancashire, UK. The carbon powder was pulverized in a ball mill for 18 h. The concentration range was 0.00 < *x* < 0.97, where *x* was the stoichiometry weight. The carbon powder was mixed in an agate mortar for 15 min to obtain a homogeneous mixture. The mixture was pressed into cylindrical pellets of 1–2 mm thickness and 5 mm diameter at 100 Kg/cm^2^ pressure for electrical measurements. The heat treatment was carried out at 473 K to eliminate the γ-AgI phase and heated to 373 K to eliminate the possible water content since AgI is hygroscopic. The mixture was heat treated at 473 K for 10 h to homogenize the distribution of the components. The sample was kept in a dry atmosphere.

Figure 4a displays the XRD patterns of AgI, Figure 4b carbon, and Figure 4c the prepared (AgI)*_x_* − C_(1−x)_ for *x* = 0.97. The corresponding tests for (AgI)_0.97_ − C_0.03_ were performed on Malvern-PANalytical X-ray diffractometer (DRX) Empyrean 2012 model, with Pixel 3D detector and Cu source (λ = 1.541 Å) at 45 kV and 40 mA; Goniometer: theta Omega 2θ and configuration platform: Spinner speed spindles at 4 rpm. The step was 0.02° and the time per step was 50 s. The representative peaks for AgI were performed with a Multi-Purpose X-ray Diffractometer PANalytical X’Pert Pro and configuration Cu Kα radiation over the angular range of 15° < 2θ < 85° at room temperature. The carbon (indexed in the American Mineralogist Crystal Structure DataBase [38]), is shown. The dashed blue lines indicate the positions of representative peaks of AgI, and the red line corresponds to carbon. The XR diffractograms of (AgI)_0.97_ − C_0.03_ mixture show that each component (AgI and C) retained its chemical identity in the mixture.

Figure 5 shows DSC measurements in the (AgI)*_x_*− C_(1−x)_ mixture, showing that AgI retains its chemical identity and exhibits the widely reported transition to the superionic state [39]. Enthalpy changes associated with the presence of the carbon phase in the sample are shown. The carbon phase absorbed some amount of the thermal energy during the phase transition of the mixture, modifying the enthalpy value.

A thorough structural characterization of the (AgI)*_x_*− C_(1−x)_ mixture is required to see the structures formed by AgI. However, this characterization is not possible using spectroscopic techniques because the AgI is degraded by radiation to sizes smaller than microns. We speculate that the cubic cells of the AgI unit of 5.03 Å are located between carbon planes separated by van der Waals forces at a distance of 5 Å.

### Phenomenological Model

The ionic conductivity ($\sigma_{i}$) can be broken down into three terms: the first, the charge of the *i*th ions ($q_{i})$; the second, the number of *i*th ions in the material per unit volume ($n_{i})$ or the fractional number of cations in interstitial sites; and the last, the mobility ($\mu_{i})$, which refers to the average velocities of the *i*th ions migrating due to an applied electric field. The expression for the $\sigma_{i}$ is given by the following:(2)$$\sigma \left(T\right)=\underset{i}{\sum }q_{i}n_{i}\mu_{i}$$

Ionic conduction involves many different ionic species moving from one point defect (Schottky or Frenkel) to another in a given crystal lattice. However, for the sake of simplicity, our system is considered to be monovalent since ionic liquids are composed of monovalent cations or anions [40]. For a crystal lattice without defects, an ideal crystal, there is no ionic conduction; therefore, point defects are necessary for this conduction (Frenkel’s defects are the most important in the ionic crystals we studied). With increasing temperature, the number of defects increases. Some phenomenological models based on point defects have been proposed to explain the temperature behavior of ionic conductivity. These models are based on the free energy density. Huberman proposed the following expression [8]:(3)$$F\left(\eta_{i}\right)=F_{0}+E\left(\eta_{i}\right)+3k_{B}T\eta_{i}\ln\left(\Gamma \right)+2k_{B}T\left\{\eta_{i}\ln(\eta_{i})+\left(1-\eta_{i}\right)\ln\left(1-\eta_{i}\right)\right\}$$The first two terms correspond to the internal energy and the last to the entropic contribution. Here, $E\left(\eta_{i}\right)$ contains two terms, one corresponding to the energy required to promote the formation of a Frenkel pair (i.e., to promote an ion to an interstitial site) and the other corresponding to the attractive interaction energy between the pairs, where η*i* is the variational parameter related to the conductivity calculated with Equation (2). The first term in $E\left(\eta_{i}\right)$ is linearly dependent on the concentration and is expressed as $U_{1}\eta_{i}$ ($U_{1}$ has energy units), and the second term as $U_{2}\eta_{i}^{2}$ ($U_{2}$ has energy units). The dimensionless factor Γ represents the ratio of the interstitial phonon frequency $\omega_{1}$ to the lattice frequency $\omega_{2}$ due to Frenkel pair formation, Γ = ω_1_/ω_2_.

By minimizing Equation (3) and evaluating it at $\eta_{i}\;$= $\bar{\eta }$, we calculated the following:(4)$$\bar{\eta }=\frac{1}{1+\Gamma^{3/2}\exp\left(\frac{\tau }{2}\left\{1-\frac{2\bar{\eta }}{\chi }\right\}\right)}$$Equation (4) is solved by numerical methods (in our case, the Newton–Raphson method was used) with at least three roots for specific given values of three dimensionless adjustable parameters: τ = *U*_1_/k_B_*T*, χ = *U*_1_/*U*_2_, and Γ. The only condition to describe a perfectly ordered crystal at *T* = 0 is that when *U*_1_ > *U*_2_ [8]. For certain values of χ and Γ, a plot of ln(η*_i_*) [proportional to ln(σ)] as a function of τ (proportional to the reciprocal of the temperature) shows two types of behavior. The first illustrates typical ion transport, i.e., no discontinuity in conductivity. The second behavior shows an exponential growth with an abrupt discontinuity in the ionic conductivity at the transition temperature. This anomaly is due to the existence of two equal minima for different values of η*_i_* in the free function (6); before *Tt*, one of them is locally stable and the other unstable, and for *T* > *Tt*, the locally stable one becomes unstable, i.e., we have a first-order transition. For the systems 0.1NaI-0.9AgI [41] and (CsHSeO_4_)_(1−*x*)_ − (KHSeO_4_)*_x_* [42], Equation (4) allowed us to fit the jump of conductivity as a function of the inverse of *T*.

To explain the behavior of the conductivity $\sigma \left(T\right)$ in all experimental temperature ranges and the first-order phase transition at *T* = *T_t_* we assumed that only some ionic carriers participated before the transition, but at *T_t_*, all participate. This phenomenology was described in [43,44]. By interpreting $\bar{\eta }$ as charge carriers, the participation of the carriers is characterized by the following probability distribution function:(5)$$P\left(\eta_{i}\right)=p\delta \left(\eta_{i}-\bar{\eta }\right)+q\delta \left(\eta_{i}\right),$$The parameter *p* is the fraction characterizing the probability that the defects have been removed, and *q* = 1 − *p* is the probability that the defects are present.

Replacing $\eta_{i}$ by $p\eta_{i}$, the new trial free energy is rewritten as follows:(6)$$F\left(\eta_{i}\right)=F_{0}+p\eta_{i}U_{1}-p^{2}\eta_{i}^{2}U_{2}+2k_{B}T\left\{p\eta_{i}\ln(p\eta_{i})+\left(1-p\eta_{i}\right)\ln\left(1-p\eta_{i}\right)+1.5p\eta_{i}\ln\left(\Gamma \right)\right\}$$

Equation (6) is the carrier density at equilibrium and describes a perfectly ordered system T=0 whenever *U_1_* > *U_2_*, under the condition *F* → *F*_0_ and $\eta_{i}$ → 0 when *T* → 0.

The corresponding state equation for the carrier density at equilibrium is as follows:(7)$$\bar{\eta }=\frac{1}{p\left\{1+\Gamma^{3/2}\exp\left(\frac{\tau }{2}\left\{1-\frac{2p\bar{\eta }}{\chi }\right\}\right)\right\}}$$
where the parameters τ, χ, Γ, and p have been defined above. Equation (7) is a transcendental equation solved with numerical (Newton–Raphson) method. This equation has three roots (η_1_, η_2_, and η_3_) and four parameters (τ, χ, Γ, and *p*). In this case, one of the three roots is locally stable (say η_1_), another is unstable (say η_2_), and the last has no acceptable physical meaning (say η_3_). The stability of the roots refers to equilibrium configurations (η = $\bar{\eta }$) that produce an absolute minimum of the trial free energy density (6) at a given temperature *T*. Similar to the case *p* = 1, Equation (7) presents two behaviors for specific values of τ, χ, Γ, and *p*. The interest is in obtaining the continuous increase in η (proportional to σ) as a function of τ (proportional to 1/*T*) to fit the experimental data of the ionic conductivity. Before the transition temperature, the locally stable root dominates (say ${\bar{\eta }}_{1}$). Above *T_t_*, the ${\bar{\eta }}_{2}$ root becomes locally stable, leading to an abrupt jump in the ionic conductivity of the system. In other words, the discontinuity in the values of $\bar{\eta }$ during the transition is due to the coexistence of two stable configurations $\bar{\eta }$ = ${\bar{\eta }}_{1}$, ${\bar{\eta }}_{2}$ with ${\bar{\eta }}_{1}$ ≠ ${\bar{\eta }}_{2}$ at the same temperature $T_{t}$ with the same minimum free energy $F\left({\bar{\eta }}_{1}\right)$ = $F\left({\bar{\eta }}_{2}\right)$.

For 0 < *p* < 1 and at the transition temperature, F($\bar{\eta }$) should have inflection points where (∂^2^F/∂*n*^2^) = 0 because F($\bar{\eta }$) has opposite concavities between $\bar{\eta }$ = ${\bar{\eta }}_{1}$, ${\bar{\eta }}_{2}$ (minima) and $\bar{\eta }$ = 1/(2*p*) (local maxima), i.e., there are inflection points. Thus, by solving (∂^2^F/∂*n*^2^) = 0, two inflection points of F(n) are obtained at
(8)$${\bar{\eta }}_{\pm }=\frac{1\pm \sqrt{1-\frac{4\chi }{\tau }}}{2p}$$These points are symmetric with respect to *n* = 1/(2*p*) from Equation (6), which has a real solution. On the other hand, since (∂F/∂*n*) = 0 [evaluated at *n* =/(2p)] at *T_t_*, Equation (7) gives the following expression in the χ:(9)$$\Gamma =\exp\left[-\frac{\tau_{t}\left(\chi -1\right)}{3\chi }\right]$$
where χ > 1. Equation (9) is independent of the value of *p*.

In summary, if the model’s parameters satisfy the following conditions, the trial free energy density (6) predicts a first-order phase transition, i.e., the density of disordered ions changes abruptly at *T_t_*:(10)$$\chi >1,\;\;\;\;\;\;\;\;\;\;\Gamma <\Gamma_{t}<1,\;\;\;\;\;\;\;\;\;\;\tau_{t}\geq 4\chi ,\;\;\;\;\;\;\;\;\;\;0<p\leq 1$$Equation (7) allowed for both the fit of the conductivity behavior of the system for *T* < *T_t_* and the jump of the ionic conductivity as a function of the inverse of temperature at *T* = *T_t_*. This equation was used to fit the AgI-KI [44] and (AgI)_(1−x)_ − (Al_2_O_3_)*_x_* nanocomposite systems [45].

One of the conditions given in Equation (10) is modified when the values for χ and Γ are chosen. There is a critical value for the probability *p*, called *p**; for the values smaller than *p**, the function (7) has only one value. For values greater than *p**, Equation (6) very well adjusts both the behavior of ionic conductivity and the abrupt jump given at the transition temperature.

With the conditions (10), the present model fit the experimental data of the ionic conductivity as a function of the temperature of the (AgI)*_x_* − C_(1−x)_ mixture for low carbon concentrations.

## 5. Conclusions

Using the conditions (10), the phenomenological model based on the free energy (6) with the probability distribution (5), the conductivity behavior for *T* < *T_t_* and its abrupt change at *T* = *T_t_* fit both behaviors well and predicted the first-order phase transitions for the (AgI)*_x_* − C_(1−x)_ mixture with low carbon concentrations (*x* = 0.01, 0.02, and 0.03).

It is observed that although carbon and AgI do not react chemically with each other, some influence of the presence of carbon on the conductivity of AgI is evident. Experimental evidence shows that by adding carbon to AgI, the conductivity behavior ceases to be the Arrhenius type. At room temperature, we found that the electrons in the carbon bands interact electrostatically with mobile silver ions, preventing their movement.

The phenomenological model is based on the formation of a vacancy, the Frenkel pair interaction, the ratio of the interstitial phonon frequency to the lattice frequency, and a probability distribution for the carrier concentration. Hoping is the mechanism used to explain the dynamics of ionic conductivity. This mechanism has been applied with good results for low carbon concentrations (*x* = 0.99, 0.98, and 0.97). However, for carbon concentrations higher than 0.03, the mechanism does not work.

The behavior of the probability *p* as a function of reduced temperature τ showed a decreasing exponential behavior. This behavior is more pronounced in this studied system than in the reported AgI and nanocomposite-based systems.

Our simulation results suggest that the curvature of the ln (σ) as a function of 1000/*T* is produced by the attractive interaction between the electrons of carbon with the Ag^+^ ions, which results in a restriction of the mobility of the silver carriers, modifying their probability of participating in the conduction as the temperature increases. The potential technological application for the system (AgI)*_x_* − C_(1−x)_ is controlling the conductivity by controlling the temperature or concentration of the mixture.

## Figures and Tables

**Figure 1 molecules-29-02491-f001:**
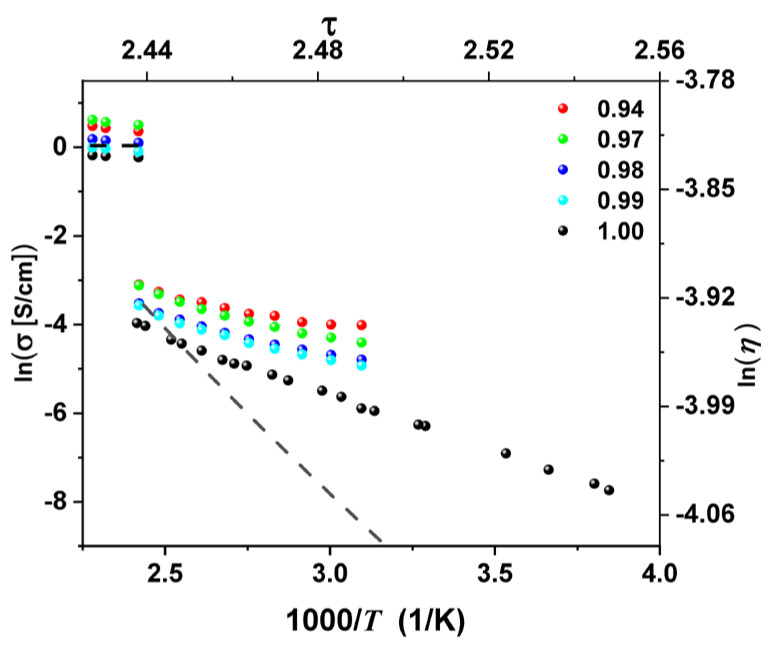
The natural logarithm of DC conductivity as a function of one thousand times the reciprocal temperature for the (AgI)*_x_* − C_(1−x)_ mixture with different carbon concentrations: pure AgI or *x* = 1.00 (black circle), *x* = 0.99 (cyan circle), *x* = 0.98 (blue circle), *x* = 0.97 (green circle), and *x* = 0.94 (red circle). The dashed line represents the best fit for the abrupt change in ionic conductivity for *x* = 0.98 using Equation (4). The other fits are not included to avoid overcrowding.

**Figure 2 molecules-29-02491-f002:**
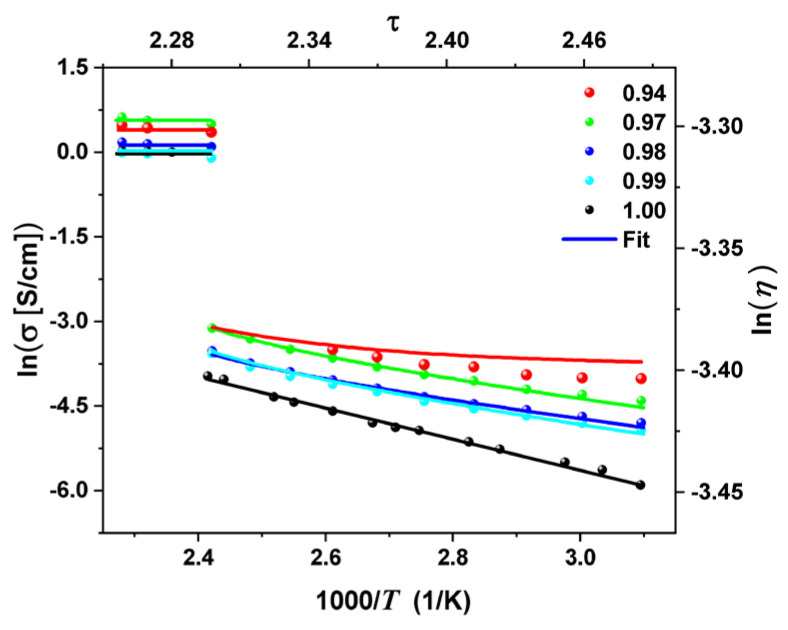
On the left, the behavior of the natural logarithm of conductivity as a function of reciprocal temperature for (AgI)*_x_* − C_(1−x)_ with *x* = 1.00 (black circle), *x* = 0.99 (cyan circle), *x* = 0.98 (blue circle), *x* = 0.97 (green circle), and *x* = 0.94 (red circle). On the right, the scale for the curves (solid lines) that best fit the experimental data using Equation (7).

**Figure 3 molecules-29-02491-f003:**
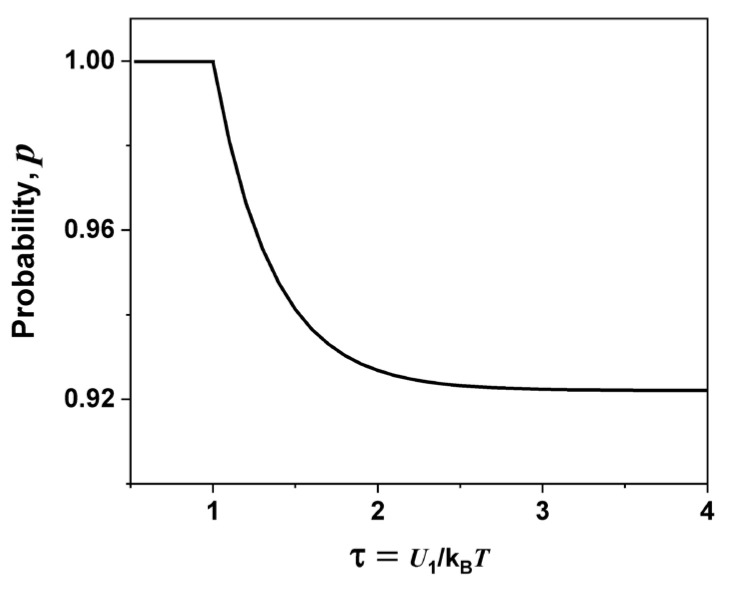
Probability as a function of τ for the (AgI)*_x_*− C_(1−x)_ mixture for a low carbon concentration (*x* = 0.98).

**Figure 4 molecules-29-02491-f004:**
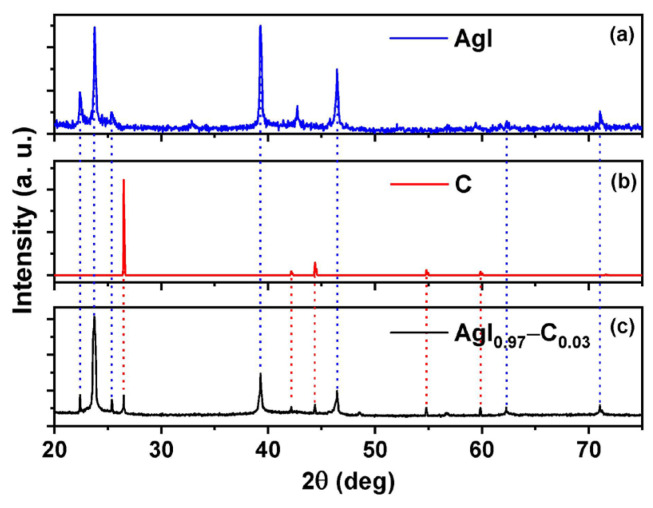
X-ray diffraction (XRD) spectra for (**a**) silver iodide, (**b**) carbon, and (**c**) (AgI)*_x_ −* C_(1−x)_. mixture, with *x* = 0.97.

**Figure 5 molecules-29-02491-f005:**
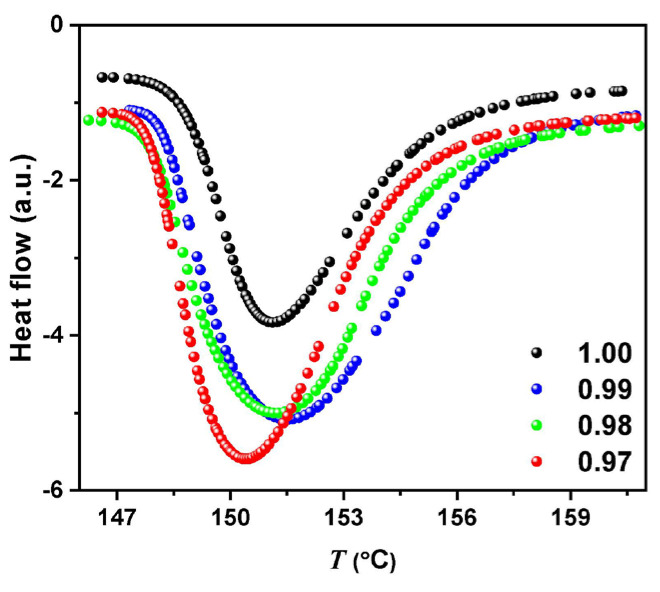
DSC measurements in (AgI)*_x_*− C_(1−x)_ mixture for *x* = 0.99, 0.98, and 0.97.

## Data Availability

The original contributions presented in the study are included in the article, further inquiries can be directed to the corresponding author/s.

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
