# Peer review of "Evidence of a Proximity Effect in a (AgI)x − C(1−x) Mixture Using a Simulation Model Based on Random Variable Theory"

_molecules, 2024, doi:10.3390/molecules29112491_

Round 1

Reviewer 1 Report (Previous Reviewer 2)

Comments and Suggestions for Authors

The paper has been greatly improved, and some contents should be clarified in the text. Note that the answer to doubt is for all readers, not reviewers.

Author Response

In the new version of our document, suggestions have been taken into account and some contents have been clarified. The content has been improved and doubts in the text have been clarified for all readers.

Reviewer 2 Report (Previous Reviewer 1)

Comments and Suggestions for Authors

Most of the questions and recommendations raised in the original version of the manuscript have been answered and taken into account. The manuscript has been improved, Now, I understood better their work. As commented previously, the emergence of mixed conduction in AgI by adding small amount of carbon graphite is an interesting behavior. I recommend its acceptance.

Some minor comments to improve the manuscript are given below.

Page 2, line 61. Carbon graphite has different structural forms -> check if the term “carbon graphite” is correct. In my opinion, “carbon” is the correct term.

The authors use the term “electrostatic correlation” or the term “correlation” (For instance, in lines 73, 76, 169). In my opinion, “electrostatic interaction” and “interaction” seem appropriate.

Author Response

In the new version of our document, suggestions have been taken into account and some contents have been clarified. The content has been improved and doubts in the text have been clarified for all readers.

To improve the manuscript, minor comments have been added to the text.

This manuscript is a resubmission of an earlier submission. The following is a list of the peer review reports and author responses from that submission.

Round 1

Reviewer 1 Report

Comments and Suggestions for Authors

The manuscript reports experimental and theoretical studies on the effect of adding carbon graphite to AgI, a well-known superionic conductor. It is reported that small amount of carbon graphite results in the emergence of a mixed conduction. This is an interesting behavior. The authors perform an analysis of the data by using a phenomenological model. It seems that the model is working. However, for the referee, some points were not sufficiently clear. In addition, the presentation is confusing. A revision is required.

The followings are some comments and questions to improve the manuscript.

1) In the section 2, the authors start the discussion by saying that the behavior can be fitted using Eqs. (3) and (7). Next move to the fitting of conductivity without explaining the meaning of the symbols, describe the materials and methods, and finally presents the model used. This order of presentation is not logical. I recommend to rearrange the order of presentation. The suggested order of presentation is: model, materials and methods, conductivity fitting and discussion.

2) If my understanding is correct, the (AgI)x–C(1−x) system is a composite. It is recommend to mention clearly this point, because, if we know that the system is a “composite”, the “proximity effect” that the authors mention make sense.

3) On the other hand, if the system is a composite and the proximity effect is playing an important role, I don’t understand why the mechanism thinking does not work for carbon graphite concentration larger than 0.03 or 3 %. Usually, composite materials can be mixed by more than 30%. Is there any explanation on why the mechanism proposed does not work for high concentration of carbon graphite mixing?

4) Related with the previous comment, it will be informative for the readers if data for carbon graphite concentration larger than 3% is shown in Fig.1 or Fig. 2.

5) I understood that “reduced temperature” is a quantity inversely proportional to the temperature. However, it seems that the term is not defined appropriately. Please define clearly the term used.

6) If my understanding is correct, the quantity eta means eta_i=(q_i)(n_i). What is the merit of using eta instead of using n? Can the electrostatic interaction between the ions (or ions and defects) extracted through the use of this quantity eta? Addition of discussion regarding this point will be informative for the readers.

7) What is the difference between eta_i and eta? This distinction is important to understand the meaning of Eq. (5). If my understanding is correct, eta_i is defined as the number of ions at the transition temperature. Is my understanding correct?

8) The authors say that (AgI)x–C(1−x) exhibit a mixed conduction. How has been confirmed this fact? How is the transference number?

9) In various parts of manuscript, the authors mention on the interaction between carbon bands and Ag ions. What does this mean? Does this mean that the Ag ion movement is affected by the layered electronic charge present in the graphite? To avoid misunderstanding additional explanation is needed.

10) Revision of English is recommended.

Comments on the Quality of English Language

A revision of English expression is recommended.

Reviewer 2 Report

Comments and Suggestions for Authors

The paper designed a model for the temperature-dependent conductivity of the AgI and C mixture based on random variable theory and revealed the so-called “proximity effect” between two compounds. The introduction and methods seemed adequate, however, there were still questions in the Results and Discussion. Therefore, I recommend that the work can be published with major revisions, and they would pay attention to the following points:

1.      The layout of the article is not very reasonable. For example, the first paragraph of Section 2 is very abrupt, and readers would prefer to first see the conductivity data. Besides, the content hierarchy of Sections 2 and 3 is not clear. By the way, the expression of equations (abbreviated or not) in whole text is not unified.

2.      If I understand correctly, Figs. 1 and 2 are identical data with different fitting equations and intervals. Therefore, the figure format should keep the same, especially for the same composition. As for “The dashed line indicates a good representative fit…”, is the fitting result in Fig. 1 really good?

3.      The dc impedance spectroscopies of AgI-C mixtures should be provided, and the ionic and electrical conductivities should be distinguished.

4.      The sentence of “the correlation between … is dominated by Ag due to the jumping mechanism” seems unreliable.

5.      Is x a mole or mass fraction? It looks very low, and how to ensure the content of C in mixtures? Is there any difference in XRD for different mixtures?

6.      The DCS results are important to analysis the performance of phase-transition materials, while the discussion of AgI-C mixtures is insufficient.

7.      Frankly speaking, the “proximity effect” is not well-explained from the conductivity results, and the assumption of “AgI lies between graphite carbon” can be further verified by molecular dynamic technology.

Comments on the Quality of English Language

Minor editing of English language required